# Kinetic Modelling of [^68^Ga]Ga-DOTA-Siglec-9 in Porcine Osteomyelitis and Soft Tissue Infections

**DOI:** 10.3390/molecules24224094

**Published:** 2019-11-13

**Authors:** Lars Jødal, Anne Roivainen, Vesa Oikonen, Sirpa Jalkanen, Søren B. Hansen, Pia Afzelius, Aage K. O. Alstrup, Ole L. Nielsen, Svend B. Jensen

**Affiliations:** 1Department of Nuclear Medicine, Aalborg University Hospital, DK-9000 Aalborg, Denmark; svbj@rn.dk; 2Turku PET Centre, Turku University Hospital, FI-20520 Turku, Finland; aroivan@utu.fi (A.R.); vesa.oikonen@utu.fi (V.O.); 3Turku PET Centre, University of Turku, FI-20520 Turku, Finland; 4MediCity Research Laboratory and Institute of Biomedicine, University of Turku, FI-20520 Turku, Finland; sirjal@utu.fi; 5Department of Nuclear Medicine and PET, Aarhus University Hospital, DK-8200 Aarhus, Denmark; soerehse@rm.dk (S.B.H.); aagealst@rm.dk (A.K.O.A.); 6North Zealand Hospital, Hillerød, Copenhagen University Hospital, DK-3400 Hillerød, Denmark; pia.maria.tullia.afzelius@regionh.dk; 7Department of Veterinary and Animal Sciences, University of Copenhagen, DK-1870 Copenhagen, Denmark; ole.lerberg.nielsen@gmail.com; 8Department of Chemistry and Biosciences, Aalborg University, DK-9100 Aalborg, Denmark

**Keywords:** kinetic analysis, Siglec-9, gallium-68, vascular adhesion protein, VAP-1, infection, inflammation, osteomyelitis, animal model, *Staphylococcus aureus*

## Abstract

Background: [^68^Ga]Ga-DOTA-Siglec-9 is a positron emission tomography (PET) radioligand for vascular adhesion protein 1 (VAP-1), a protein involved in leukocyte trafficking. The tracer facilitates the imaging of inflammation and infection. Here, we studied the pharmacokinetic modelling of [^68^Ga]Ga-DOTA-Siglec-9 in osteomyelitis and soft tissue infections in pigs. Methods: Eight pigs with osteomyelitis and soft tissue infections in the right hind limb were dynamically PET scanned for 60 min along with arterial blood sampling. The fraction of radioactivity in the blood accounted for by the parent tracer was evaluated with radio-high-performance liquid chromatography. One- and two-tissue compartment models were used for pharmacokinetic evaluation. Post-mortem soft tissue samples from one pig were analysed with anti-VAP-1 immunofluorescence. In each analysis, the animal’s non-infected left hind limb was used as a control. Results: Tracer uptake was elevated in soft tissue infections but remained low in osteomyelitis. The kinetics of [^68^Ga]Ga-DOTA-Siglec-9 followed a reversible 2-tissue compartment model. The tracer metabolized quickly; however, taking this into account, produced more ambiguous results. Infected soft tissue samples showed endothelial cell surface expression of the Siglec-9 receptor VAP-1. Conclusion: The kinetics of [^68^Ga]Ga-DOTA-Siglec-9 uptake in porcine soft tissue infections are best described by the 2-tissue compartment model.

## 1. Introduction

Infections causing localized lesions in the body, e.g., osteomyelitis, cannot always be successfully treated by the systemic administration of antibiotics; instead, surgical intervention may be necessary, which requires knowledge of the site(s) of infection. Assuming the availability of a suitable tracer, positron emission tomography (PET) is well suited for the task. However, it has proven difficult to find a tracer with both high and specific uptake in infected tissue.

The current standard tracer used for PET imaging of infectious and inflammatory diseases is the fluorine-18 labelled glucose analogue 2-deoxy-2-[^18^F]-fluoro-d-glucose ([^18^F]FDG) [1]. Generally, [^18^F]FDG shows high uptake in sites of infection and inflammation, but the uptake is not specific: [^18^F]FDG accumulates in all cells that use glucose as an energy source, including not only activated immune cells but also rapidly proliferating cancer cells, brain tissue and working muscle tissue.

Leukocytes (white blood cells, WBC) are natural seekers of infection. Leukocytes labelled with ^111^In or ^99m^Tc have been widely used for scintigraphy imaging of infection, but the labelling procedure is time consuming and requires withdrawing 30–40 mL of blood from the patient [2]. A simpler alternative is anti-granulocyte monoclonal antibodies (anti-G-mAbs) labelled with ^99m^Tc. However, while the isotopes ^111^In and ^99m^Tc are suitable for gamma camera imaging, they cannot be used for PET imaging. This is a technical disadvantage, as the counting efficiency is typically an order of magnitude higher for the PET scanner than for the gamma camera, and the spatial resolution is superior. Labelling of leukocytes with [^18^F]FDG for PET imaging has been investigated, but the labelling efficiency is lower, and [^18^F]FDG elutes from the leukocytes over time [3]. Furthermore, the molar activity of labelled leukocytes will be many orders of magnitude higher than in the patient as a whole, for which reason the short-range β^+^ radiation (positrons before annihilation) can result in damaging self-irradiation to [^18^F]FDG-labelled leukocytes [4].

The relationship between time and physical decay must also be noted. Injected leukocytes take hours or more to accumulate at the infection site, and imaging can be performed 3–4 h after injection at the earliest [2]. These delays are acceptable with ^111^In (T_½_ = 2.8 days) and ^99m^Tc (T_½_ = 6 h), but in relation to PET imaging, the shorter physical half-lives of the two most widely used isotopes, ^18^F (T_½_ = 110 min) and ^68^Ga (T_½_ = 68 min), make long uptake times problematic.

Instead of labelled leukocytes, a tracer directly related to the processes causing leukocyte accumulation in infected tissues is being sought. Such a tracer would potentially enable faster imaging, and at the same time, the handling of blood products from the patient would be avoided. A relevant target molecule is vascular adhesion protein 1 (VAP-1), currently known as a primary amine oxidase (AOC3, EC 1.4.3.21), which is acting both as an adhesion molecule and as a regulatory enzyme in the process of leukocyte binding to the endothelium of blood vessels in infected and inflamed tissue [5,6]. Under normal conditions, VAP-1 is stored in intracellular granules. Upon an inflammatory stimulus, VAP-1 is rapidly translocated to the endothelial cell surface, where it is readily accessible to intravenously administered PET ligands. 

As reviewed in [7], ^68^Ga-labelled synthetic peptides that bind to VAP-1 have been investigated in different experimental infection and inflammation models, and some have shown a better distinction than [^18^F]FDG between cancer and inflammation. Overall, the review considers VAP-1 to be an optimal target for imaging of inflammation. One of the reviewed tracers, [^68^Ga]Ga-DOTA-Siglec-9, is currently in a phase I clinical trial (trial no. NCT03755245).

It may be noted that VAP-1 imaging will not distinguish between infection and inflammation, but because VAP-1 is involved in leukocyte extravasation, it is directly linked to the body’s natural response to infection. Imaging of VAP-1 expression may even be used to study this process, especially if the kinetics of the VAP-1 imaging tracer are known.

Sialic acid-binding immunoglobulin-like lectin 9 (Siglec-9) is a natural ligand for VAP-1 and is expressed on monocytes and neutrophils [8,9]. The PET tracer [^68^Ga]Ga-DOTA-Siglec-9 contains a modified fraction of the full Siglec-9 protein and has been found to accumulate in infected tissues [10,11] as well as inflammation sites [8,11,12,13] and certain tumours [8]. However, the kinetics of [^68^Ga]Ga-DOTA-Siglec-9 have been investigated only in a single study that addressed inflammation [12]; thus studies on the kinetics of this tracer in infections are lacking.

In earlier studies, we investigated the PET imaging of infection in a porcine osteomyelitis model. In this model, *Staphylococcus aureus* (*S. aureus*) bacteria were injected into the right femoral artery of juvenile pigs to haematogenously induce osteomyelitis in the right hind limb without trauma to the bone [14,15,16]. In a number of animals, both osteomyelitis and soft tissue infections were induced in the limb, allowing us to compare the uptake in these two types of tissues. Each animal was scanned with a series of radioactive tracers within the same day, with planned delays to allow one tracer to physically decay before the next was injected [17].

The present study investigates the kinetics of [^68^Ga]Ga-DOTA-Siglec-9 in this porcine osteomyelitis model, including a comparison of the uptake in osteomyelitis and in soft tissue infections.

## 2. Results

The characteristics of the eight pigs in the study are summarized in Table 1.

### 2.1. Visual Uptake

Visually, [^68^Ga]Ga-DOTA-Siglec-9 showed elevated uptake in inflamed and infected soft tissues, while not being prominent in infected bone (osteomyelitis). As an example, a coronal view of the uptake around the femur in pig no. 25 is shown in Figure 1. 

At late time points, the decay-corrected PET signal was markedly reduced in all tissues outside the bladder, indicating that uptake of the tracer is reversible rather than irreversible (see Figure 2).

### 2.2. VAP-1 Expression Studied by Immunofluorescence (IF) 

To analyse the expression of VAP-1 in infectious/inflammatory foci and to determine whether VAP-1 is translocated from intracellular granules to the endothelial cell surface, pig no. 25 was administered anti-VAP-1 antibody shortly before euthanasia, as described in the Methods section. The tissue samples were stained either with anti-VAP-1 + secondary antibody or with secondary antibody alone. Together, this pair of stainings allowed us to detect both the total VAP-1 pool and the part of the pool reached by the intravenously injected antibody; presumably only the latter pool is accessible to the imaging peptide. It can be noted that even though Siglec-9 and anti-VAP-1 both bind to VAP-1, their binding sites are different [9]; thus, they do not compete for binding.

VAP-1 was highly expressed in abscess-associated vasculature and a portion of this pool was detected with the intravenously injected antibody (Figure 3A,B). While VAP-1 is intracellularly expressed in certain vessels of normal soft tissue, no signal was detected with the intravenously administered antibody (Figure 3C,D). It thus appears that VAP-1 is accessible to blood-circulating antibodies only in the infected/inflammatory tissue, which can be seen as an indication that the translocation of VAP-1 to the cell surface is inflammation-induced. 

### 2.3. Protein Binding of Radioactivity 

The plasma protein binding results are shown in Figure 4. Overall, protein binding seemed to vary relatively little over the time of scanning (the late measurement in pig no. 10 deviates, but may also have the largest uncertainty due to the low level of radioactivity remaining at the late time point). A constant fraction of protein binding reflects an equilibrium between the protein-bound and non-bound tracer, allowing the plasma to be treated as a single compartment. Partly for this reason, and partly because we lack protein data for the rest of the pigs, we did not include protein binding in further analysis.

### 2.4. Analysis of the Parent Tracer Fraction 

The parent tracer fraction (fraction of radioactivity originating from [^68^Ga]Ga-DOTA-Siglec-9) rapidly decreased during the investigated time. A sample fraction curve for the parent tracer is shown in Figure 5. All fraction curves are shown in Appendix A.

### 2.5. VOIs and Model Fits

In the 9 scans, a total of 24 VOI pairs were drawn: 10 VOI pairs were drawn in bone and 14 VOI pairs were drawn in soft tissue positions (counting the VOIs in pig no. 26 twice, as this pig was scanned twice). In pig no. 24, no infectious lesions were identified. In several pigs, bone lesions in the small pedal bones had to be excluded from VOI drawing and analysis, as the resolution of the PET scans would result in a partial volume effect that would be too high for the results to be robust. The studied lesions are summarized in Table 2 and Table 3. The individual VOI volumes (cm^3^) can be found in Appendix A.

For each VOI, the PET data were fitted with the models shown in Figure 6. Using pig no. 9 as an example, the model fits based on uncorrected and corrected input function are shown in Figure 7 and Figure 8, respectively. All model fits are shown in Appendix A.

Many of the Patlak plots showed signs of non-linearity and/or horizontal fits, indicating that uptake was reversible rather than irreversible. The Patlak plots for pig no. 9 are shown in Figure 9, and all Patlak plots are shown in Appendix A.

Plotting the fitted parameters from the corrected vs. uncorrected input function (plots not shown) revealed that the main difference was a highly elevated ratio *k*_3_/*k*_4_ if the corrected input function was used. A high value of *k*_3_/*k*_4_ indicates that the tracer leaves the second compartment only slowly, i.e., a nearly irreversible uptake. Accordingly, the Patlak plots based on the corrected input function were closer to linearity than the Patlak plots based on the uncorrected input function (cf. Figure 9 and Appendix A).

### 2.6. The Corrected Akaike Information Criterion (AIC_c_)

The AIC_c_ values favoured the rev2TCM model, whether the input function was the uncorrected or the corrected one. Specifically, when the uncorrected input function was used, rev2TCM gave the lowest AIC_c_ in 18 of the 24 investigated (18/24) infection VOIs and 17/24 control VOIs. Using the corrected input function, the corresponding numbers were 17/24 and 20/24, respectively. See Appendix A. 

### 2.7. The Volume of Distribution in Tissue (V_T_)

Based on the AIC_c_ results, only the *V_T_* results from the rev2TCM were considered. 

The volume of distribution in tissue (*V_T_*) from the VOIs in the infected (right hind limb) and the control (left hind limb) positions are compared in Figure 10. Generally, the *V_T_* results for bone infections (osteomyelitis) were close to the identity line, i.e., similar uptake in infection and control tissue. For the soft tissue infections, most points were above the identity line, indicating higher uptake in the infected soft tissues than in the corresponding control tissues, although this result was less clear when the corrected input function was used.

Based on the modelling results from uncorrected input function, the *V_T_* differences between infection VOIs and control VOIs were statistically significant in the soft tissue infections (*p* < 0.0001), but not in the bone infections (*p* = 0.83). More precisely, for soft tissue VOI pairs (*n* = 14), the mean difference *∆V_T_ = V_T,_*_infection_
*− V_T,_*_control_ was 0.18 mL/cm^3^, with 95% limits of agreement (LOA) from 0.12 to 0.25 mL/cm^3^. For bone VOI pairs (*n* = 10), the mean difference was 0.005 mL/cm^3^, with 95% LOA from −0.050 to +0.062 mL/cm^3^. No signs of non-normal distributions were found (*p* > 0.1 for both soft tissue and bone infections).

### 2.8. Correlation Between Standardized Uptake Value (SUV) and V_T_

Only *V_T_* values based on the uncorrected input functions were considered. The Pearson correlation coefficient was *r* = 0.54 (*r*^2^ = 0.29, *n* = 48, *p <* 0.0001), indicating a moderate correlation. A plot is shown in Appendix A.

Often, SUV is not evaluated as an absolute value but relative to a reference tissue. If relative rather than absolute values of *V_T_* and SUV were plotted, then correlation rose to *r* = 0.66 (*r*^2^ = 0.43, *n* = 24 pairs, *p* = 0.0005). A plot is shown in Figure 11. 

## 3. Discussion

In this study, we investigated the uptake of [^68^Ga]Ga-DOTA-Siglec-9 in juvenile pigs with localized *S. aureus* infections, including both bone infection (osteomyelitis) and soft tissue infection. 

As in previous studies [10,13], the tracer was found to have an affinity for infected tissues. Siglec-9 is known to bind to VAP-1, and IF staining was consistent with the expression of VAP-1 on cell surfaces in the infected soft tissue.

Kinetic modelling showed that [^68^Ga]Ga-DOTA-Siglec-9 had reversible uptake kinetics, which could be described with a two-tissue compartment model (i.e., rev2TCM from Figure 6). Visually, the models with fewer parameters provided good fits in many cases (see the 1TCM and irr2TCM curves in Figure 7 and Figure 8); however, the 1TCM curves provided poor fits for the initial part of many curves, and the Patlak plots revealed that models with irreversible uptake such as irr2TCM could not be generally applied (cf. Figure 9). Likewise, the analysis by the AIC_c_ values (cf. Appendix A) favoured rev2TCM in the large majority of cases. 

Based on these results, the following discussion will assume rev2TCM as the chosen model. Physiologically, rev2TCM indicates that the tracer is taken up by the tissue (first compartment), and then the tracer binds to receptors or otherwise changes status in the tissue (second compartment), but with the possibility of unbinding/changing back (reversible model).

### 3.1. Using the Corrected or the Uncorrected Input Function 

The radio-HPLC analysis of blood samples revealed a rapid decrease in the parent tracer fraction (Figure 5). However, modelling showed no clear advantage of using a corrected input function. In most fits, the visual difference between using the uncorrected or corrected input function was only minor (compare Figure 7 and Figure 8). 

Physiologically, fitting with the uncorrected input function assumes that any radioactive metabolites have the same uptake kinetics as the parent tracer. In contrast, fitting with the corrected input function assumes that the radioactive metabolites have no uptake at all. The real situation is probably somewhere in between. 

Mathematically, the inclusion of such a correction has the disadvantage of including a source of uncertainty, which becomes especially important in a case such as the present where the correction in the late part of the scans was considerable (cf. Figure 5). 

The volume of distribution (*V_T_*) measures how concentrated the tracer is in tissue relative to plasma (input), cf. Equation (4) in the Material and Methods section. As shown by comparing the scales in Figure 10, the calculated values of *V_T_* depend markedly on the selection of the uncorrected or corrected input function. This dependency is a simple consequence of the math. The corrected input function is by definition only some fraction of the total, uncorrected input function (cf. Figure 5). A given radioactivity concentration in the tissue will be relatively higher when compared to a low number (the corrected input function) than when compared to a higher number (the uncorrected input function).

In summary, our data showed a rapid decrease in the parent tracer (cf. Figure 5), but unfortunately the modelling did not allow us to distinguish between the PET signal from the parent tracer (i.e., [^68^Ga]Ga-DOTA-Siglec-9) and the PET signal from other radioactive species formed in vivo (i.e., all metabolites containing the ^68^Ga isotope, possibly including free ^68^Ga). 

Consequently we reason that use of the corrected input function will lead to increased uncertainty of *V_T_* due to possible errors in the measurement of parent tracer fraction. When the uncorrected input function is used, this pitfall is avoided, although at the cost of risking over-simplification. Pragmatically, the following discussion will therefore focus on the results found with the uncorrected input functions, bearing in mind that the physiological reality is likely more complicated than the model. Further studies investigating the nature of the radioactive products of [^68^Ga]Ga-DOTA-Siglec-9 formed in vivo are warranted.

### 3.2. The Volume of Distribution V_T_

The *V_T_* for [^68^Ga]Ga-DOTA-Siglec-9 in soft tissue infections was found to be higher than that of the corresponding control sites, (red circles in Figure 10, left part). In bone infections (osteomyelitis), however, *V_T_* was not significantly different between the infection and control sites (blue squares in Figure 10, left part).

Retamal et al. [12] used the same tracer to study lung inflammation, also performing kinetic modelling with the model called rev2TCM in our terminology (with no mention of correction for parent tracer fraction), and found that the model described the time–activity curves well. The study also included protein binding, which was found to be constant over time, at a level of approximately 20% in healthy pigs and approximately 50% in the inflamed pigs. Our results (cf. Figure 4) are mostly in accordance with the first of these numbers, which could reflect that from a systemic perspective, a local infection in a single limb is more similar to a healthy pig than to a pig with severe inflammation in both lungs. Retamal et al. found increased uptake of [^68^Ga]Ga-DOTA-Siglec-9 in inflamed lungs, which is consistent with our results on uptake in soft tissue infection.

In summary, [^68^Ga]Ga-DOTA-Siglec-9 shows increased uptake in infected (and inflamed) soft tissue compared with control tissue; however, this study fails to demonstrate elevated uptake in infected bone (osteomyelitis). These quantitative results correspond to the visual impression of the sample image in Figure 1, where increased uptake is clear in infected soft tissue but not in the infected bone (cf. with pig no. 25 in Table 2 and Table 3). Note, however, that this comparison is partly qualitative. The determination of *V_T_* on a reliable absolute scale will depend on improved knowledge on the nature of the radioactive metabolite products of [^68^Ga]Ga-DOTA-Siglec-9. The moderate correlation between SUV and *V_T_* indicates that the volume of distribution gives information that is not just a complicated version of the SUV. 

In a previous study [18] (in part performed on the same animals), we found only a small increase of blood perfusion in osteomyelitis lesions, while blood perfusion was considerably increased in soft tissue infections. As speculated in that study, an ineffective vascular response to infection may lead to too few leukocytes reaching the infected bone, in part explaining why osteomyelitis is difficult for the body to fight. Similarly, despite the uptake of [^68^Ga]Ga-DOTA-Siglec-9 in infected tissue, imperfect perfusion can impair effective uptake, which may explain the difference in results for soft tissue infections and osteomyelitis.

### 3.3. Scan after the Injection of “Cold” DOTA-Siglec-9

Pig no. 26 was scanned twice, and the second scan was performed after the injection of 5 mg of unlabelled DOTA-Siglec-9 peptide, which was intended to block the VAP-1 receptors. The uptake curves for both scans are shown in Appendix A. For the infected lymph node, the bolus passage and initial uptake show differences between the first and the second scan, but otherwise the two sets of curves are very similar. Quantitative distribution volumes are listed in Appendix A. Rather than the expected decrease from receptor blocking, both lesions in pig no. 26 show an approximately 20% increase in *V_T_* from the first scan to the second when the uncorrected input function is used. With only two lesions in one pig, however, it is difficult to draw conclusions.

Using the corrected input function, Appendix A shows pronounced *V_T_* differences (still increases) for the two lesions in pig no. 26, but we are hesitant to draw conclusions from these results, as they may reflect the sensitivity of *V_T_* to the correction (cf. Section 3.1). In addition, pig no. 26 unfortunately showed the lowest parent tracer fractions (cf. Appendix A) and therefore had the largest sensitivity to possible errors in the correction of the input function.

### 3.4. Limitations

A porcine model has the advantage over, e.g., a murine model that the physical sizes involved in both surgery and scans are larger, but the disadvantage is that the cost per animal is higher. Accordingly, this study is limited by a relatively small number of animals.

The non-traumatic osteomyelitis protocol has the advantage that it is a very realistic model of haematogenous osteomyelitis (in humans and animals) and infections were reasonably limited to the right hind limb, but the numbers and locations of infections varied among the animals. However, within the individual pig, the other hind limb could be used as a control, and for soft tissue, the uptake measured as the volume of distribution (*V_T_*) in infected versus control sites showed quite clear differences (Figure 10). As already noted, the study does not claim to report *V_T_* on a robust absolute scale, which would also require data on protein binding in all animals.

A scanner replacement resulted in the last three pigs (no. 24–26) being scanned on a PET/CT scanner of a different brand and a newer generation than the initial scanner, enhancing the spatial resolution of the PET images. This allowed refined VOI drawing in these pigs, as reflected in the generally smaller VOI sizes for these animals (cf. Appendix A). More precisely drawn VOIs could reduce a possible partial volume effect, in which case uptake would be expected to be more pronounced in pigs no. 24–26 than in the pigs scanned on the initial scanner. However, the results for *V_T_* based on the uncorrected input function (cf. Appendix A) do not indicate any pronounced effect. Using the corrected input function, pigs no. 25 and 26 do have high *V_T_* values, but these values reflect the pronounced correction for the (possibly artificially low) parent tracer fractions in these pigs (cf. Appendix A), while an effect of voxel size should also be reflected in *V_T_* calculated using the uncorrected input function.

Another potential source of error may be that the scans were performed on anaesthetized pigs, as the anaesthetic may affect blood flow and hence the kinetics of the tracer. However, it was not practically feasible to scan awake pigs. Propofol was chosen because it provides relatively uniform and safe anaesthesia over many hours.

The use of penicillin in the animals may have reduced the extent of the studied infections, but has previously been found to provide a better balance between the successful development of osteomyelitis and the avoidance of systemic infections requiring euthanasia of the animal [23]. The use of opioids can have immunosuppressant effects, but so can pain reactions, which are reduced by the pain-killer effect; buprenorphine was chosen because it has shown weaker immune effects than morphine and fentanyl [24]. As the target molecule VAP-1 for the tracer is not related to the infecting agent but is instead part of the natural immune system response, there is no reason to expect direct interference between [^68^Ga]Ga-DOTA-Siglec-9 and the antibiotic pharmaceuticals (penicillin and buprenorphine).

## 4. Materials and Methods

### 4.1. Porcine Infection Protocol

The animal protocol was approved by the Danish Animal Experimental Board, journals no. 2012-15-2934-00123 (original approval) and 2017-15-0201-01239 (renewed approval, no substantial changes to the protocol), and all procedures followed the European Directive 2010/63/EU on the protection of animals used for scientific purposes. The protocol for the haematogenous induction of osteomyelitis in domestic pigs has been detailed elsewhere [23]; for general background, see [14,15,16]. 

Briefly, *S. aureus* was inoculated into the right hind limb of juvenile (19–25 kg) Danish Landrace–Yorkshire crossbred female pigs. The pigs were pre-acclimated for at least one week, during which they were housed in groups in boxes with restricted access to food (Dia plus FI, DLG, Copenhagen, Denmark) and *ad libitum* access to tap water. The temperature was 20–24 °C, the humidity was 45–65%, and there were 12 h of darkness and 12 h of light in the stables. The pigs came from a specific pathogen free (SPF) herd and were clinically examined by a veterinarian before inoculation. The pigs were fasted for approximately 16 h before premedication with midazolam and Stresnil and propofol anaesthesia. After inoculation, the pigs were individually housed. The *S. aureus* used was the porcine strain S54F9 [25], and 8000 to 30,000 CFU/kg (colony forming units per kg body weight) were inoculated. To selectively infect the right hind limb, bacteria were injected into the right femoral artery. To further reduce the possibility of systemic infection, the pigs were administered penicillin (10,000 EI/kg) at the onset of the first clinical signs of disease; for such juvenile pigs this dosage has previously been shown to allow the development of osteomyelitis while minimizing cases of systemic infection [23]. To reduce pain in the days until euthanasia, the animals were treated every 8 h with buprenorphine (Temgesic; 0.3 to 0.9 mg intramuscularly, depending on clinical symptoms). Osteomyelitis was allowed to develop for approximately one week, after which the pig was scanned with PET and computed tomography (PET/CT, see below) and euthanized. If a pig reached predefined humane endpoints [23] before this time, it was euthanized (and not scanned). 

Although the protocol was originally designed for inducing osteomyelitis [17], several pigs also developed soft tissue infections in the inoculated hind limb, typically related to bone infection or the site of inoculation. This turned out to be an advantage, as it allowed us to compare osteomyelitis and soft tissue infections. Some of the pigs also developed lung abscesses. However, these abscesses were outside the field-of-view of the dynamic PET scans and are therefore not included in this kinetics study.

### 4.2. Animals and Lesions

This study includes eight pigs dynamically scanned with PET/CT using [^68^Ga]Ga-DOTA-Siglec-9 (details below). The characteristics of these pigs are summarized in Table 1. These eight pigs are a subset of the animals from the overall osteomyelitis project but represent all pigs in the project scanned with this tracer. 

Before euthanasia, each pig was also PET/CT scanned with [^18^F]FDG; these scans have been reported earlier and found [^18^F]FDG to be a sensitive (but unspecific) marker of *S. aureus* infection [17,19,26]. Infectious lesions were identified on [^18^F]FDG PET/CT scans. As part of the post-euthanasia analysis, selected lesions were verified by necropsy to be suppurative and to be caused by the inoculated *S. aureus* strain, S54F9. 

For the identified lesions, volumes of interest (VOIs) for PET data analysis were drawn on the [^68^Ga]Ga-DOTA-Siglec-9 PET/CT scans using Carimas 2.9 software (Turku PET Centre, www.turkupetcentre.fi/carimas). For osteomyelitis lesions, the VOI drawing was based on the CT part of the scan, while for soft tissue lesions the VOI drawing was guided by the PET scan.

For each lesion, a reference VOI was drawn in the anatomically corresponding position in the left, non-infected limb.

### 4.3. Radiochemistry

The radioactive labelling of DOTA-Siglec-9 has previously been discussed and described in detail [22]. The [^68^Ga]Ga-DOTA-Siglec-9 radiosynthesis method called method 3 in the reference was applied in this study. 

In summary, ^68^Ga was eluted from a ^68^Ge/^68^Ga generator (GalliaPharm, Eckert & Ziegler, Berlin, Germany), trapped on a cation-exchange cartridge (Strata-XC 33 u Polymeric Strong, Phenomenex, Værløse, Denmark), and eluted from the cartridge with an acidified acetone solution. The pH was adjusted using HCl (0.1 M in metal-free water), and acetone was removed by heating. A solution of DOTA-Siglec-9 peptide in metal-free water was added, and ^68^Ga incorporation took place. Water was added to the mixture, which was then run through a preconditioned C-18 Sep-Pak cartridge to trap [^68^Ga]Ga-DOTA-Siglec-9. The product [^68^Ga]Ga-DOTA-Siglec-9 was released from the cartridge with ethanol and diluted with saline. After this process, the product was ready for injection.

After 25 min, a 62% non-decay-corrected (ndc) yield of the product was obtained. The [^68^Ga]Ga-DOTA-Siglec-9 was found by a radio HPLC system to be more than 98% radiochemically pure, and the specific radioactivity was approximately 35 MBq/nmol. Representative radio-HPLC chromatographs are shown in Appendix A.

### 4.4. Dynamic PET Scans

Before scanning, each pig was anaesthetized with propofol, and catheters were implanted in the jugular vein and carotid artery [17]. After an initial CT scan, the pig was dynamically PET scanned for 60 min in 23 time frames: 8 × 15 s, 4 × 30 s, 2 × 60 s, 2 × 120 s, 4 × 300 s, and 3 × 600 s. The [^68^Ga]Ga-DOTA-Siglec-9 tracer was injected into the jugular vein at the start of the PET scan. The tracer activities are shown in Table 1. All of these scans were performed at the Department of Nuclear Medicine, Aalborg University Hospital.

Pigs no. 6–23 were scanned on a GE VCT Discovery 64 PET/CT scanner (GE Healthcare, Chicago, IL, USA). The scan field covered 15 cm in the axial direction and was positioned over the pelvis and the hind limbs. The images were reconstructed with an ordered subset expectation maximization (OSEM) algorithm (3D Vue Point, GE). The reconstruction parameters were 2 iterations, 28 subsets, a 128 × 128 matrix in 47 slices, a 5.5 × 5.5 × 3.3 mm^3^ voxel size, and a 6 mm Gaussian filter.

Due to scanner replacement, pigs no. 24–26 were scanned on a different scanner than the previous pigs. Pigs no. 24–26 were scanned on a Siemens Biograph mCT (Siemens, Erlangen, Germany) with time-of-flight (TOF) detection. The scan field covered 22 cm in the axial direction, positioned over the pelvis and the hind limbs. The images were reconstructed with an OSEM algorithm without using the resolution recovery option (setting “Iterative + TOF”). The reconstruction parameters were 3 iterations, 21 subsets, a 400 × 400 matrix in 1.02 × 1.02 × 2.03 mm^3^ voxels, and a 3 mm Gaussian filter.

On both scanners, image reconstruction included decay-correction to the start of scanning and attenuation-correction based on the CT scan.

### 4.5. Blood Samples

An arterial blood sample was drawn before the tracer was injected (zero-sample). During the dynamic PET scan, 27 blood samples were drawn. All blood samples were drawn manually from the carotid artery, and the precise time (seconds) of each sample was recorded. Time zero was the time of injection, which was also the scan start time.

In pigs no. 6–10, the blood samples were drawn every 5 s for 50 s (10 samples), at 60, 80, 100, 120, and 150 s post-injection (p.i., 5 samples), and at 3, 4, 5, 6, 8, 10, 15, 20, 30, 40, 55, and 70 min p.i. (12 samples), i.e., 27 blood samples. Samples for analysis of the fraction of radioactivity originating from the parent tracer (rather than from radioactive metabolite products or free gallium) were drawn at 2, 5, 10, 15, 25, 35, 50, and 70 min p.i.

In pigs no. 22–26, the blood sample timing was slightly optimized. Blood samples were drawn every 5 s for 40 s (8 samples), at 50, 60, 80, 100, 120, and 150 s p.i. (6 samples), and at 3, 4, 5, 6, 8, 10, 15, 20, 25, 30, 40, 50, and 60 min p.i. (13 samples), i.e., 27 blood samples. Blood samples for analysis of the parent tracer fraction were drawn at 2, 5, 10, 15, 25, 40, and 60 min p.i.

As noted in Table 1, pig no. 26 was scanned twice, and 5 mg of “cold” (unlabelled) DOTA-Siglec-9 was injected before the second scan. Blood sampling was independently performed for the two scans.

Plasma samples were obtained by collecting the supernatant after the centrifugation of whole blood samples. Aliquots of the samples were counted in a calibrated Wizard 2480 gamma counter (PerkinElmer, Turku, Finland) with an energy window from 450 to 1200 keV. The counting results were converted to decay-corrected radioactivity concentrations (Bq/mL at the time of injection).

The plasma samples for analyses of the parent tracer fraction were denatured by thoroughly mixing 0.5 mL plasma with 0.5 mL acetonitrile, after which the mix was centrifuged (approximately 1 min, 10,000 rpm) to accelerate the precipitation of the plasma proteins. An aliquot was collected for HPLC analysis (see below). The radioactivity of the precipitate was determined with the Wizard 2480 gamma counter. Protein binding was calculated as
(1)$$\mathrm{protein}\;\mathrm{binding}=\frac{\mathrm{precipitate}\;\mathrm{radioactivity}/0.5\;\mathrm{mL}}{\mathrm{plasma}\;\mathrm{radioactivity}\;\mathrm{concentration}}\times 100\%,$$
using decay-corrected activities.

For the determination of parent tracer fractions in a sample, 0.2 mL of the supernatant was diluted with 0.8 mm water; this dilution was run through an HPLC with a fractionation collector. The fraction collector was set up with a 20 s delay to compensate for delays in the system. Twenty fractions of 45 s each were collected and counted in the Wizard 2480 gamma counter.

### 4.6. Tissue Samples and Immunofluorescence (IF)

As a proof of concept, the surface expression of VAP-1 in infected tissue was tested by an augmented protocol in one pig.

Approximately 10 min before euthanasia, pig no. 25 was administered 10 mg of VAP-1-binding antibody as an intravenous (i.v.) injection (10 mL injected liquid). The antibody was 1B2, a mouse IgM against human VAP-1 that also recognizes porcine VAP-1 [27]. The i.v. injection allowed 1B2 to bind to the VAP-1 expressed on cell surfaces, but not to the VAP-1 within intact cells.

After euthanasia of the pig, soft tissue samples were collected from acutely inflamed areas (phlegmon/early abscess) located peripheral to the osteomyelitis in the distal right femur and from similarly positioned non-inflamed areas in the left hind limb. All samples were embedded in a cryo-compound and frozen in petroleum spirit (VWR, Søborg, Denmark, cat. no. 87125.320) cooled with dry ice. The tissues were stored at −80 °C until use.

Immunofluorescence (IF) analysis was performed on 5 µm thick frozen sections of these samples. The first of two serial sections from the inflamed area (right limb) were stained with anti-VAP-1 mAb (1B2) or a class-matched negative control antibody, 7C7 (10 µg/mL; 1 h at room temperature) and then with a secondary antibody (Alexa 555-goat-anti-mouse IgM 1:100, Southern Biotech 1020-32; 30 min at room temperature), followed by nuclear staining with Hoechst 1:10,000 in PBS for 5 min, Thermo Scientific 6249, Waltham, MA, USA). The combined IF signal thus represented the VAP-1 within the cytoplasm as well as the VAP-1 expressed on cell surfaces. The second section was stained only with the secondary antibody; thus, the IF signal represented only the VAP-1 accessible to the i.v. injected antibody. A pair of sections from the left limb was similarly stained to represent the corresponding signals from non-infected tissue.

### 4.7. Input Function and Metabolite Correction

For each dynamic PET/CT scan, the decay-corrected plasma samples were used as a basic input function. 

For metabolite correction, the parent tracer fraction data were fitted with a Hill-type function:(2)$$f\left(t\right)\;=\;1\;-\;\frac{\left(1-a\right)t^{b}}{c\;+\;t^{b}},$$
where *t* is the sampling time (seconds since injection). The function starts at *f*(0) = 1 (thus assuming no metabolism before injection) and has an asymptotic value f(∞) = *a*. The parameters *a*, *b* and *c* were fitted for each injection of [^68^Ga]Ga-DOTA-Siglec-9. 

In the following text, the *uncorrected input function* denotes the total radioactivity concentration (decay-corrected Bq/mL) from plasma samples, and the *corrected input function* denotes the fraction *f*(*t*) multiplied by the uncorrected input function.

As blood samples were taken from the carotid artery (a short distance from the heart), while the PET data were acquired over the hind end of the pig (a longer distance from the heart), the PET data were delayed by some seconds relative to the blood plasma data. To correct for these delays, an offset to the plasma time stamps was determined for each pig using the method described in [19]. The largest of these corrections was 8 s.

### 4.8. Kinetic Modelling

Data were modelled using the three kinetic models shown in Figure 6. This was performed twice for each model: once with the uncorrected input function and once with the corrected input function. The modelling was performed using software available on the Turku PET Centre website (fit2k for 1TCM, fit3k for irr2TCM, fit4k for rev2TCM) [28]. 

Data points were weighted based on time frame length (*L*) and decay:(3)$$w\;=\;L\;\times \;\exp\left(-\lambda t\right)\;=\;L\;\times \;0.5^{t/T_{½}}$$

This weighting scheme mirrors the overall count statistics of the decay-corrected PET data. Unlike weights based on counts in a VOI, the weights from Equation (3) do not in themselves contain noise. See reference [19] for a more detailed discussion of this weighting scheme.

For in vivo imaging, the volume of distribution in tissue (*V_T_*) is defined as the ratio of tissue concentration to input concentration at a time when a steady-state has been reached [29], i.e.,
(4)$$V_{T}\;=\;\frac{\mathrm{tissue}\;\mathrm{concentration}}{\mathrm{input}\;\mathrm{concentration}}\;(\mathrm{at}\;\mathrm{steady}\text{-}\mathrm{state})$$

The measurement of tissue concentration in Bq/cm^3^ (from the PET scan) and input concentration in Bq/mL (from plasma samples in the gamma counter) results in mL/cm^3^ as the unit of *V_T_*.

For a given model with reversible uptake, the relationship between *V_T_* and the model parameters can be theoretically calculated. In the cases of the 1TCM and the rev2TCM models, these relationships are [29]:(5)$$1\mathrm{TCM}:\;V_{T}\;=\;\frac{K_{1}}{k_{2}}$$
(6)$$\mathrm{rev}2\mathrm{CTM}:\;\;V_{T}\;=\;\frac{K_{1}}{k_{2}}\cdot \left(1+\frac{k_{3}}{k_{4}}\right)$$

For models with irreversible uptake, such as irr2TCM, a steady-state is never reached and *V_T_* is not defined. Mathematically, the tracer input concentration will be continually decreasing, while the uptake in the irreversible compartment will monotonically increase, and over time, the fraction in Equation (4) will diverge instead of converging.

The parent tracer fraction appeared to decrease faster in pig no. 26 (both scans) than in the other pigs, and was close to zero after approximately 30 min post-injection (see Appendix A). The corrected input function in this case would therefore be close to zero (expectedly with high percentage errors) at late time points, which could lead to unreliable estimation of *V_T_* = tissue concentration/input concentration. For these reasons, modelling of pig no. 26 with the corrected input function was performed using only the data within the interval 0–30 min rather than the full interval 0–60 min.

### 4.9. Evaluation

In addition to visual inspections of the fits, the three models were compared using the corrected Akaike information criterion (AIC_c_), which favours a good fit, but penalizes the use of excessive fitting parameters [30,31]. The absolute value of AIC_c_ depends on both the data and the model, but for a given data set, the lowest value of AIC_c_ indicates the most favourable model.

Furthermore, Patlak plots [32,33] were calculated to help determine whether uptake was reversible or irreversible. A linear Patlak plot (after an equilibration time) is a sign of irreversible uptake, while a system with only reversible uptake will result in a non-linear Patlak plot that eventually approaches a constant value. The Patlak plot is directly calculated from the data, without the assumption of any specific uptake model. The Patlak plots were based on the data from 10 to 60 min (10–30 min in pig no. 26 with corrected input function).

As a measure not requiring modelling, standardized uptake values (SUV) were also calculated, and the correlation between SUV and the volume of distribution *V_T_* was determined. The SUV calculation was based on the time interval 10 to 30 min, chosen as a time interval after the passage of the bolus peak, but not so late that a considerable part of the tracer with reversible binding (i.e., tracer not remaining in the tissue indefinitely) would have left the tissue yet. 

### 4.10. Statistics

To compare the infected and control VOIs (i.e., right vs. left), a two-tailed paired *t*-test was used, with a significance level of *p* < 0.05. The normality of the differences was tested with the Shapiro–Wilk W test. The correlation between the volume of distribution *V_T_* and SUV was defined using Pearson analysis. Statistics were calculated with StatsDirect version 3.1.14 (www.statsdirect.com).

## 5. Conclusions

Using the VAP-1-targeted leukocyte ligand Siglec-9, the immunofluorescence analysis of infected tissue samples indicated that VAP-1 was expressed on the cell surfaces in infected tissue, while surface VAP-1 was not observed in non-infected tissue.

The uptake kinetics of [^68^Ga]Ga-DOTA-Siglec-9 with localized infection in pigs were found to be well described with a reversible 2-tissue compartment model, similar to the model used by Retamal et al. [12] in a study of severe lung inflammation, also in anaesthetized pigs. 

We found that the parent tracer fraction decreased relatively rapidly, but despite this finding, we were unable to demonstrate an advantage of taking tracer metabolism into account in the analysis. More detailed analyses of the radioactive species occurring after the i.v. injection of [^68^Ga]Ga-DOTA-Siglec-9 in the body are warranted.

The [^68^Ga]Ga-DOTA-Siglec-9 uptake, evaluated as the volume of distribution, showed affinity to infection in soft tissue; however, no increased uptake in bone infections (osteomyelitis) could be demonstrated. This difference may be related to a previous report that found infected soft tissue to be more highly perfused than infected bone tissue (osteomyelitis).

## Figures and Tables

**Figure 1 molecules-24-04094-f001:**
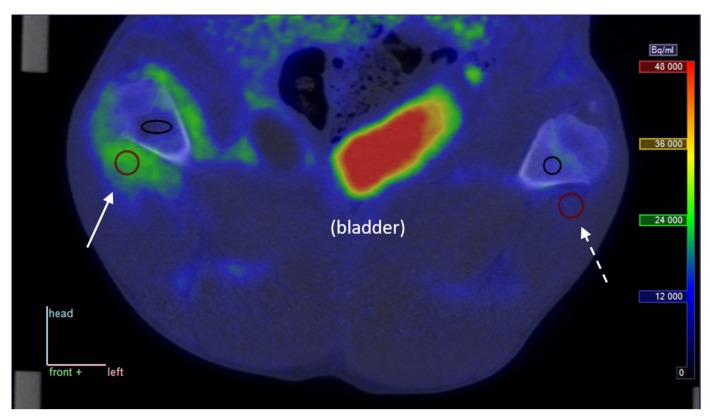
Representative PET/CT image from pig no. 25. The pig was lying supine with roughly upward-pointing limbs, and the view is coronal, seen from the ventral side of the animal (“front” = opposite of back is in the direction of the reader); the top of the image is in the direction of the animal’s head. Full arrow: Uptake in phlegmon/early abscess at the right distal femur. Dotted arrow: Similar non-infected tissue for comparison. Red circles show sections of volumes of interest (VOIs) in soft tissue, black ellipse/circle show VOIs in bone tissue. The PET image represents the summed data from 15 to 30 min post-injection (p.i.). The tissue samples presented in Figure 3 were taken from positions approximately corresponding to the soft tissue VOIs.

**Figure 2 molecules-24-04094-f002:**
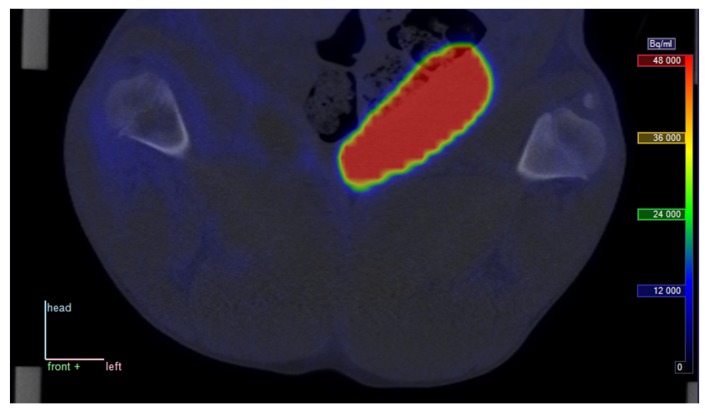
Late PET/CT image from pig no. 25. Static scan with [^68^Ga]Ga-DOTA-Siglec-9 at approximately 70 min p.i.; position and colour scale correspond to those in Figure 1 (radioactivity is corrected for physical decay).

**Figure 3 molecules-24-04094-f003:**
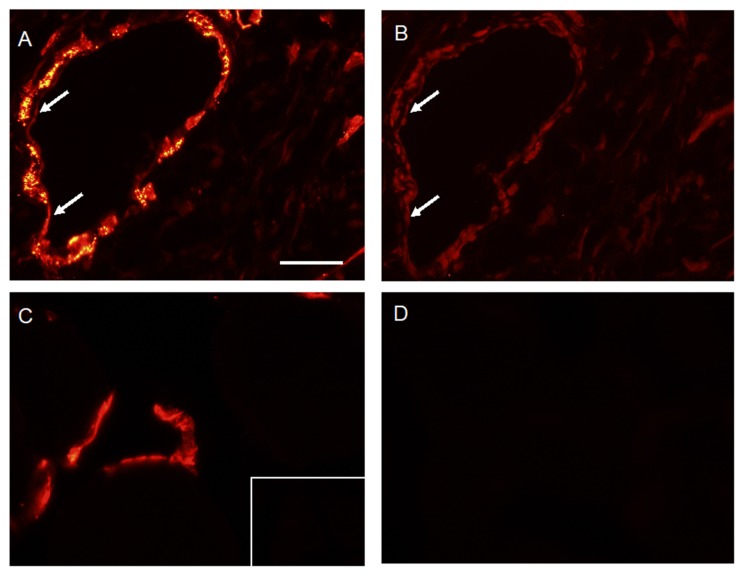
IF staining of infected and non-infected tissue. Stained samples from pig no. 25. **A** and **B** are stains of serial sections from phlegmon/early abscess at the right distal femur. For comparison, **C** and **D** are stains of a non-infected soft tissue sample from an anatomically corresponding position in the left limb of the same animal. (**A**) Staining with anti-VAP-1 and the secondary antibody showed high intracellular and surface expression of VAP-1 in vessels. The scale bar is 50 µm. (**B**) Staining only with the secondary antibody showed VAP-1 surface expression alone and thus indicated that a part of the VAP-1 pool is translocated to the endothelial cell surface (arrow). A signal is also detected on the abluminal side, indicating that the antibody gained access via inter-endothelial junctions. (**C**) Anti-VAP-1 staining followed by the secondary antibody. Negative control staining is shown in the inset. (**D**) Staining only with the secondary antibody showed no VAP-1 surface expression in the non-inflamed tissue.

**Figure 4 molecules-24-04094-f004:**
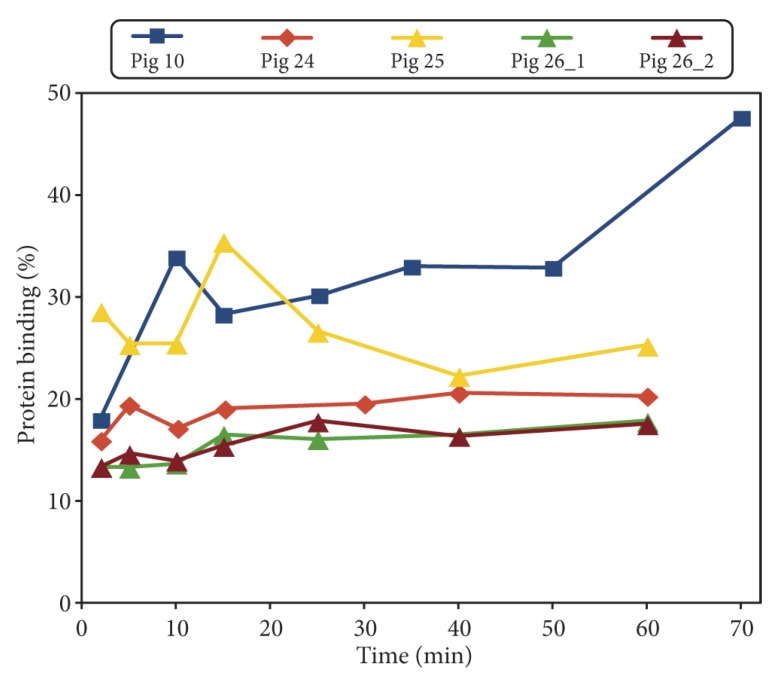
Plasma protein binding of [^68^Ga]Ga-DOTA-Siglec-9 in the pigs. Percentage of binding as a function of minutes after tracer injection. For pig no. 10, the data are a summary of previously published results [22].

**Figure 5 molecules-24-04094-f005:**
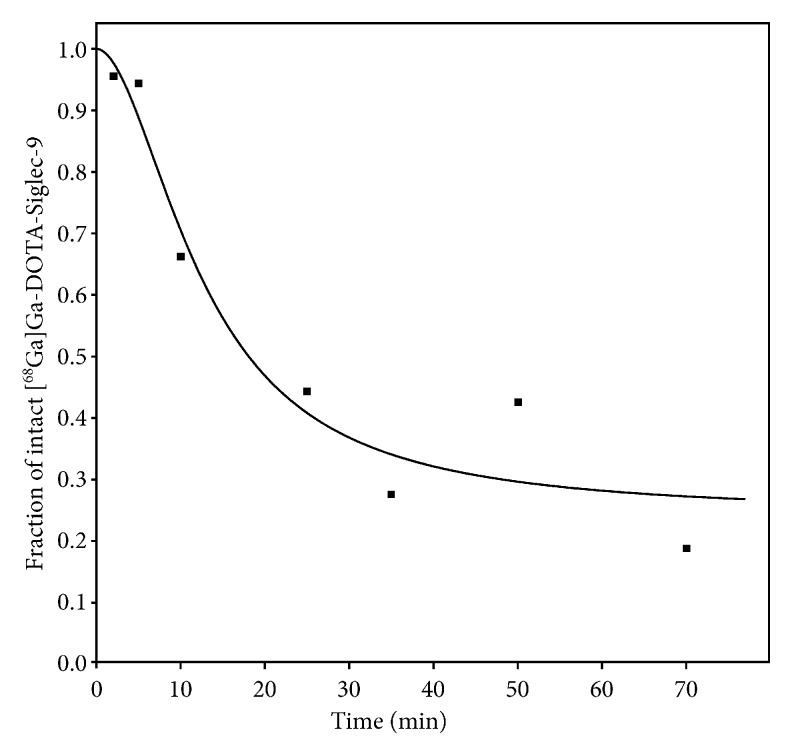
Fraction of radioactivity from intact [^68^Ga]Ga-DOTA-Siglec-9 (the parent tracer fraction) in arterial plasma as a function of time. Data points and curve fits (Equation (2)) are from pig no. 9 as a representative example.

**Figure 6 molecules-24-04094-f006:**
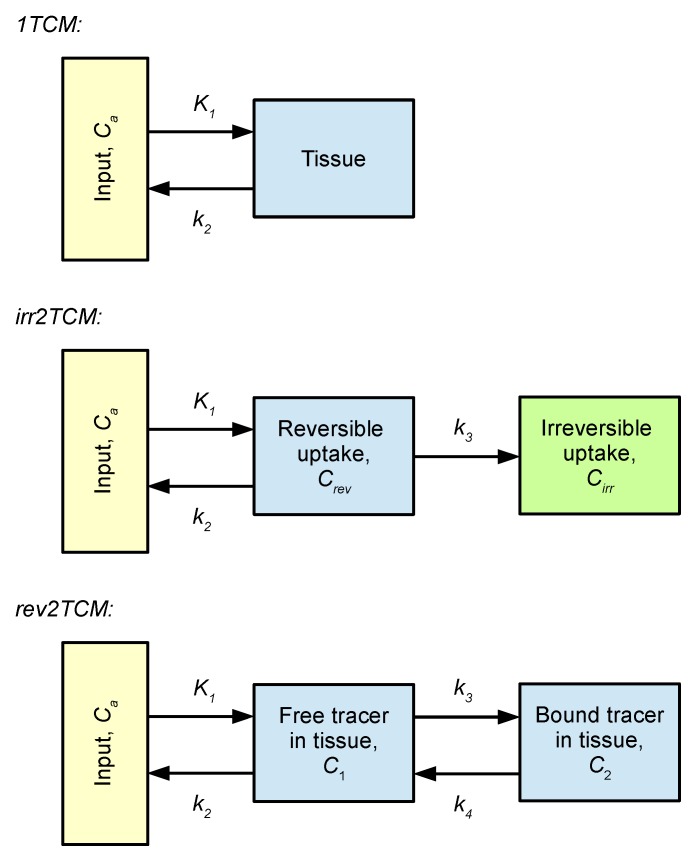
Kinetic models applied for the quantification of [^68^Ga]Ga-DOTA-Siglec-9 uptake. From top to bottom: 1-tissue compartment model (1TCM), 2-tissue compartment model with irreversible uptake (irr2TCM), and 2TCM with reversible uptake (rev2TCM). The rate constants were fitted as *K*_1_ (unit mL/min/cm^3^ or mL/min/100 cm^3^), the ratio *K*_1_/*k*_2_ (unit cm^3^/mL), *k*_3_ (unit min^−1^), and the ratio *k*_3_/*k*_4_ (no unit). Additionally, the blood fraction in tissue *V_b_* was fitted.

**Figure 7 molecules-24-04094-f007:**
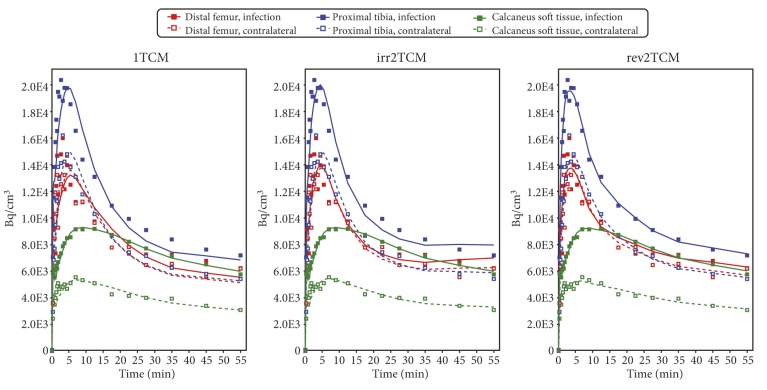
Model fits for pig no. 9 with the *uncorrected* input function. Infection data are from osteomyelitis lesions in the distal femur and proximal tibia and from a soft tissue infection at the calcaneus, all in the animal’s right hind limb. Control data are from anatomically corresponding positions in the noninfected left hind limb. All data were modelled with each of the three kinetic models shown in Figure 6.

**Figure 8 molecules-24-04094-f008:**
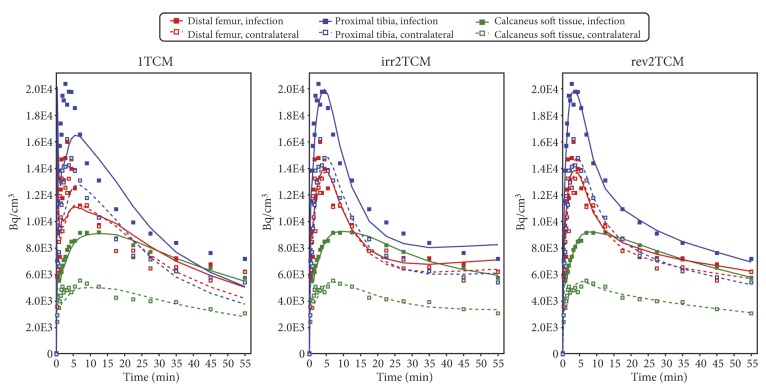
Model fits for pig no. 9 with the *corrected* input function. Same PET data as in Figure 7, different input function. The plots are very similar to those in Figure 7, but not identical; visual differences from Figure 7 are most notable for the early part of 1TCM.

**Figure 9 molecules-24-04094-f009:**
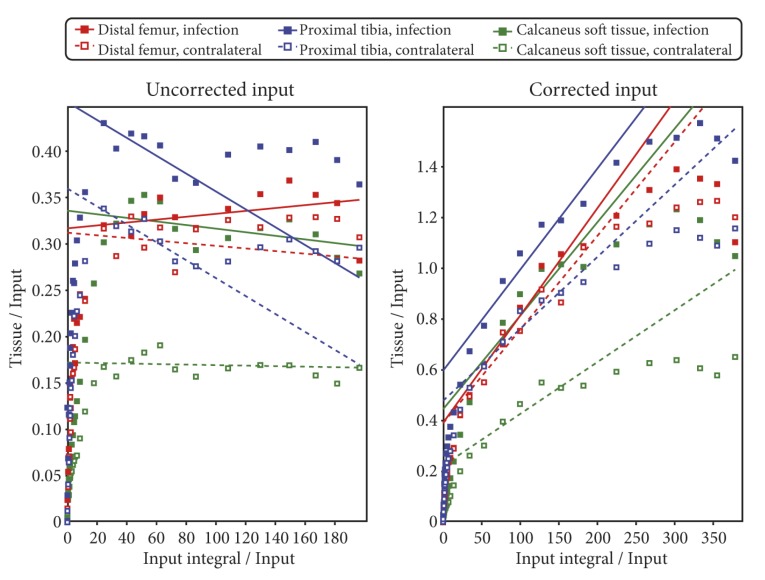
Patlak plots for pig no. 9. **Left**: Based on the uncorrected input function (data as in Figure 7). **Right**: Based on the corrected input function (data as in Figure 8). After the early points, irreversible uptake will be characterized by a linear Patlak plot with a positive slope, while nonlinearity or horizontal slope are signs of reversible uptake.

**Figure 10 molecules-24-04094-f010:**
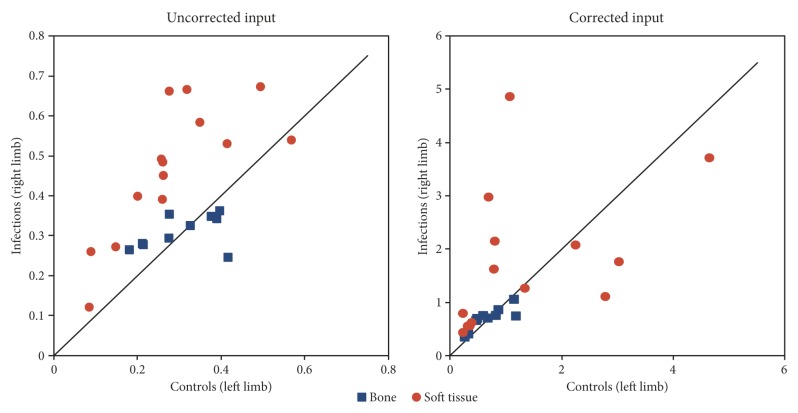
Volume of distribution in infected versus control tissues. Data are the *V_T_* from Equation (6) in infection VOIs and control VOIs. Blue squares are bone VOIs, and red circles are soft tissue VOIs. **Left**: *V_T_* from fit with the uncorrected input function. **Right**: *V_T_* from fit with the corrected input function. Positions on the line correspond to equal values for the infection and control positions. The underlying data of the plots can be found in Appendix A. See Discussion for a comparison of the scales.

**Figure 11 molecules-24-04094-f011:**
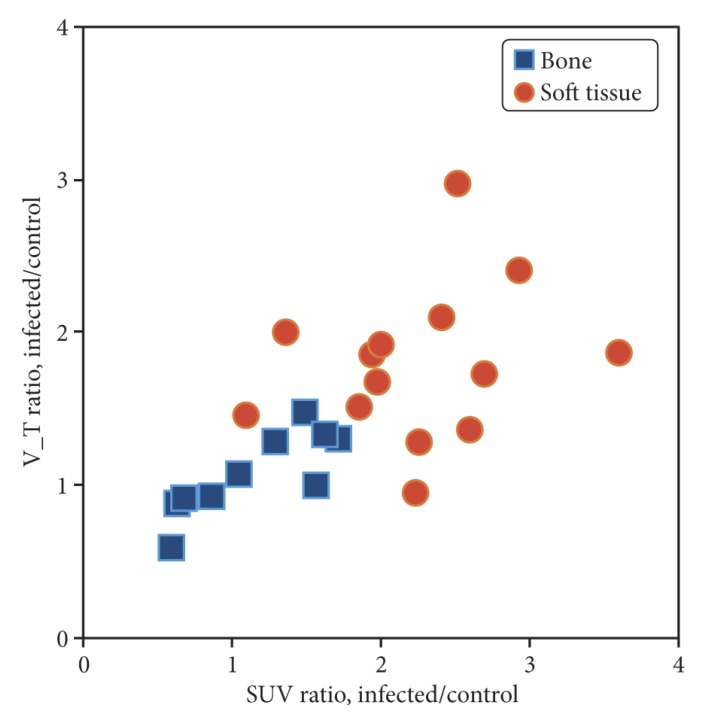
Correlation between relative distribution volume and relative SUV. In the plot *V*_*T*-ratio_ = *V*_*T*,infected_ / *V*_*T*,control_ and SUV_ratio_ = SUV_infected_ / SUV_control_, where for each infection VOI, the control VOI is at the corresponding location in the non-infected limb of the same animal VOI (see main text).

**Table 1 molecules-24-04094-t001:** Characteristics of the pigs in the study.

Pig No.*	Body Weight (kg)	Days from Inoculation to Scan	Injected Radioactivity (MBq)
6	22.0	7	178
9	23.0	8	175
10	22.5	7	191
22	19.0	9	208
23	19.0	9	150
24	20.0	8	126
25	20.5	8	253
26 ^†^	24.5	8	227
			360

* The numbering follows the order in the overall osteomyelitis project described in the main text. Pigs no. 6, 9, and 10 have been described regarding other tracers in previous publications: [18,19] (same numbers) and [20] (where A–E corresponds to numbers 6–10). All pigs in the table except no. 24 were also included in [21] (with different numbering). Blood sample data on [^68^Ga]Ga-DOTA-Siglec-9 in pig no. 10 have been described in [22]. ^†^ Pig no. 26 was scanned twice with [^68^Ga]Ga-DOTA-Siglec-9. The second scan was performed 4.6 h (approximately 4 physical half-lives of ^68^Ga) after the first scan and with a higher level of injected radioactivity. Prior to the second scan, 5 mg of unlabelled (“cold”) DOTA-Siglec-9 peptide in 10 mL had been slowly injected intravenously (i.v.) into the pig to block receptors—see main text.

**Table 2 molecules-24-04094-t002:** Investigated osteomyelitis lesions, right hind limb.

Region	Pigs No.
Distal femur	6, 9, 22, 23, 25
Proximal tibia	6, 9, 10, 23
Distal tibia	10

**Table 3 molecules-24-04094-t003:** Investigated soft tissue infections sites, right hind limb.

Region	Pigs No.
Soft tissue at distal femur	22, 23, 25^†^
Soft tissue at distal tibia	10
Abscess at calcaneus	9
Soft tissue infection/abscess in the tissue plantar to lateral intermediary phalanxes	10, 25, 26*
Abscess at inoculation site (capsule)	22, 23
Abscess at inoculation site (centre)	22
Abscess in superficial popliteus lymph node	26*

* Pig no. 26 was scanned twice; between the two scans, 5 mg of unlabelled (“cold”) DOTA-Siglec-9 peptide was injected into the animal. ^†^ See also Figure 3A,B.

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
