# Peer review of "Kinetic Modelling of [68Ga]Ga-DOTA-Siglec-9 in Porcine Osteomyelitis and Soft Tissue Infections"

_molecules, 2019, doi:10.3390/molecules24224094_

Round 1

Reviewer 1 Report

General remarks

The recognition of hidden sites of inflammation belongs to the most frequently asked questions not only related to diseases of the skeletal system but also to pre-stages of diverse other pathological processes. The Siglec lectin subfamily includes transmembrane proteins which can bind sugar and recognize sialic acid. Expressed in granulocytes and monocytes Siglec or fractions of the protein can act as ligand of the vascular adhesion protein-1 (VAP-1) involved in inflammatory reactions.  

The authors promote the use of a modified fraction of a Siglec-9 protein linked to [68Ga]Ga-DOTA for identification of  membrane presentation of VAP-1 with positron emission tomography (PET)/computer tomography (CT). The tracer is applied to a model of haematogenously induced osteomyelitis in domestic pigs. The tomographic evaluation of the inflammatory/infectious foci was performed within 7 to 9 days after injection of Staphylococcus aureus.

The authors based their model on profound experience with the porcine model of haematogenous induction of osteomyelitis and associated inflammatory alterations of soft tissue adjacent to the bones investigated. Infection is induced in the right hind limb, intraarterially, observing the animals 7 to 9 days after infection. The animal experiments were finished after last PET/CT investigation or, alternatively, with euthanasia and postmortem immunofluorescence labelling of soft tissue sections. During the experiments the animals got analgesics to prevent pain. Application of Staphylococcus aureus was accompanied by penicillin protection for prevention of sepsis and general spreading of the bacteria in the body. Infectious foci are investigated at femur and tibia at different places of their occurrence as well as at adjacent soft tissue foci at calcaneus in the right hind limb. The PET data and radioHPLC data obtained for the applied tracer in blood samples were employed for pharmacokinetic modelling. The experimental protocols have been approved by Danish Experimental  Animal Board according to the European guidelines on animal care.

The authors discuss critically the limitations of their model which is restricted in the possibilities of statistical analysis because of the  small number of animals, the non-uniform distribution of infectious foci which, moreover, are found predominantly in the soft tissue adjacent to the bone than as affection of the bone itself. Furthermore, for investigation of some of the animals a new PET camera was used, possibly with altered geometry of the detectors and field of view, i.e. that the camera changed during the experimental series.

The authors, however, restrict also their conclusions in pharmacokinetic modelling on the evaluation of single animal monitoring, the test of the suitability of 1- and 2-tissue compartment models, PET- and immunofluorescence identification of the VAP-1 protein in soft tissue and displacement with cold tracer. They give very skilled information on their pharmacodynamic and pharmacokinetic PET approaches. Finally, especially the model investigations are done with background of currently starting clinical phase I studies. At least, these investigations of the non-rodent, mammalian model of inflammatory processes can provide valuable preclinical data for the PET approach in man and the proof of concept for characterization of inflammatory processes with Siglec-9 based radiotracers. Finally, the article provides also some educational potential also for readers which are relatively new in the field.

Minor remarks:

Page 4, Fig. 1 and legend of Figure 1: The coordinate system given at the left side of the Figure is not easy to understand related to the information that the animal was investigated in supine position. The abscissa indicates “front” of the animal at the bottom of the figure. Should that not be the backside in supine position?

        Line 4, last word, typos: interest instead of interes

Page 13: It is a little bit confusing that the absence of metabolite data is discussed under point 3.1 (last paragraph) as an advantage and in point 3.2.(third paragraph) as a disadvantage.

Page 13, point 3.2, and Conclusions (page 19): The authors discussed in the Introduction predominantly the importance of the Siglec-9 tracer in comparison with the [18F]FDG, effective for detection of infectious loci. It would be interesting, also in relation to other models and other inflammatory markers (for instance with PK11195) used by the authors, which position they give [68Ga]Ga-DOTA-Siglec-9 radiotracers used in the context of inflammation.

Page 15, Material and Methods, lines 1 and 2: Please, show which animals belong to which protocol (journal no.)?

Author Response

Thanks for the elaborate and positive evaluation. Our comment to the specific remarks follow below.

Page 4, Fig. 1, “front” direction:

The direction “front” is meant to point out of the screen/paper (+), towards the reader. This would be the front (anterior) side of a human being lying on the back. To make this more clear, we have added this text to the figure legend:

“seen from the ventral side of the animal (“front” = opposite of back is in the direction of the reader)”

The typo has been corrected.

Page 13, subsections 3.1 and 3.2, advantage or disadvantage:

Our intention is not to call the absence of metabolic data an advantage, but in 3.1 we find that including these data (theoretically) increases the risk of errors while it (empirically) makes very little difference for the visual fit, even if it influences the values of the fitted parameters. For these reasons we take the pragmatic approach not to use the data, but are aware that this may represent an oversimplification that could influence the absolute values of the volume of distribution in the tissue (VT). Accordingly, we express caution in subsection 3.2, comparing the VT values only on a relative scale while we refrain from evaluating the absolute values.

3.2 and Conclusions, context of inflammation instead of infection:

The infection protocol used in the study has the advantage of closely mimicking haematogenous infection, but the disadvantage that the site(s) of infection changes somewhat among the animals. This put demands on the resources needed for necropsy and biopsies, and focus was therefore on infection rather than inflammation. Rather than augment our discussion on topics we have only briefly touched experimentally, we prefer to keep our present focus on infection. However, information of the tracer in relation to inflammation can be found in references [8,11-13] of the paper as it is.

Page 15, two protocol approvals mentioned in 4.1 – which animal belongs to which protocol? Actually, the two numbers represent original and prolonged approval, not two different protocols. The text has been augmented to make this clear.

Reviewer 2 Report

The manuscript by Jødal et al, entitled "Kintetic modeling of [68Ga]Ga-DOTA-Siglec-9 in porcine osteomyelitis and soft tissue infections" describes the evaluation of the pharmacokinetic modelling of a very promising radioligand for imaging infectious and inflammatory diseases. 

The manuscript is in general terms of good quality and gathers quite a significant amount of work. The conclusions are well supported by experimental data and the literature references are appropriate. The paper might be suitable for publication, although a few points need to be considered:

1- For the determination of the parent tracer fraction (free fraction), the plasma samples were denatured by adding acetonitrile. Wouldn't this process affect the protein-compound interaction? Wouldn't be more appropriate to use the ultracentrifugation method? this way, the plasma proteins would be retained in the filter and the peptide and metabolites would be collected in the filtrate. 

2- Wouldn't be more appropriate to correct the input functions by the "free fraction" instead of total plasma radioactivity? If so, would this have a significant impact in terms of volume of distribution? 

3- It would be convenient to include a representative radio-HPLC chromatogram in the supplementary material to show the metabolites that are present in plasma apart from the parent tracer. 

Author Response

We thank the reviewer for the positive evaluation. Below we respond to the specific comments.

1, denaturation by acetonitrile. We chose acetonitrile as a standard method for denaturation. While ultrafiltration avoids adding chemicals (thereby excluding interaction with these chemicals), there is instead a risk of interaction with the filter that can be important for material present only in tracer doses, as is the case in this study. So while we acknowledge that acenotrile is not guaranteed to be completely interaction-free, we consider it as robust as ultrafiltration in this case.

2, influence of the free fraction / parent tracer fraction on the volume of distribution. There may be two different concepts here: non-protein-bound (free) activity and activity belonging to Siglec-9 rather than to a metabolite. But the influence on volume of distribution (VT) will be of the same kind: If the input function is reduced by a fraction, then VT will be correspondingly increased, I.e., multiplied by 1/fraction; see equation (4) and our discussion in section 3.1.

So yes, the absolute value of VT can be markedly influenced by this choice (compare the VT results for uncorrected and corrected input function in supplementary table S4). And yes, if we had had sufficiently solid data for these effects, it would be appropriate to take this into account.

However, for a given pig (a given scan), the same input function will be used for all VOIs, roughly corresponding to a common correction factor for all VOIs. Thus the RELATIVE values of V_T in infection vs. control (or in bone tissue vs. soft tissue) will not depend much on this choice. Thus, we are safe in comparing within the individual pigs, while care should be taken if comparisons are made between pigs.

3, inclusion of a radio-HPLC chromotogram in the supplementary material. A sample radio-HPLC chromatogram has been included as Supplementary Figure S11, with a short reference in the methods section of the main paper (subsection 4.3).